# Novel Poly(Caprolactone)/Epoxy Blends by Additive Manufacturing

**DOI:** 10.3390/ma13040819

**Published:** 2020-02-11

**Authors:** Andrea Dorigato, Daniele Rigotti, Alessandro Pegoretti

**Affiliations:** Department of Industrial Engineering and INSTM research unit, University of Trento, 38123 Trento, Italy; daniele.rigotti-1@unitn.it (D.R.); alessandro.pegoretti@unitn.it (A.P.)

**Keywords:** epoxy, poly(caprolactone), blends, 3D printing, self-healing, fracture toughness

## Abstract

The aim of this work was the development of a thermoplastic/thermosetting combined system with a novel production technique. A poly(caprolactone) (PCL) structure has been designed and produced by fused filament fabrication, and impregnated with an epoxy matrix. The mechanical properties, fracture toughness, and thermal healing capacities of this blend (EP-PCL(3D)) were compared with those of a conventional melt mixed poly(caprolactone)/epoxy blend (EP-PCL). The fine dispersion of the PCL domains within the epoxy in the EP-PCL samples was responsible of a noticeable toughening effect, while in the EP-PCL(3D) structure the two phases showed an independent behavior, and fracture propagation in the epoxy was followed by the progressive yielding of the PCL domains. This peculiar behavior of EP-PCL(3D) system allowed the PCL phase to express its full potential as energy absorber under impact conditions. Optical microscope images on the fracture surfaces of the EP-PCL(3D) samples revealed that during fracture toughness tests the crack mainly propagated within the epoxy phase, while PCL contributed to energy absorption through plastic deformation. Due to the selected PCL concentration in the blends (35 vol %) and to the discrepancy between the mechanical properties of the constituents, the healing efficiency values of the two systems were rather limited.

## 1. Introduction

In the last few decades, epoxy resins—thanks to their high mechanical properties, thermal and chemical stability, good processability, and adhesion to different substrates—have found wide application in different fields [1,2,3]. However, in many technological applications, the mechanical properties of epoxy resin are not able to fulfill all the technical requirements imposed during the service conditions [4,5,6]. This is the reason why these thermosetting polymers are often coupled with other materials, in order to produce fiber reinforced polymers (FRPs) and polymer blends [7]. In FRPs, different kinds of long or short inorganic fibers (i.e., glass, carbon, Kevlar, basalt, etc.) can be added to the epoxy [8,9], while in polymer blends epoxy can be mixed with thermoplastic or thermosetting matrices [10].

The high mechanical, thermal, and chemical properties achievable in epoxy/thermoplastic blends make them suitable for different applications, such as matrices for composites, materials with self-healing and shape memory characteristics, and aerospace components with good strength and fracture toughness [11]. In the literature, it can be found that several thermoplastic polymers were used for blending epoxy resin, such as polycarbonate (PC), poly(caprolactone) (PCL), and poly(ethylene terephthalate) (PET), and different kinds of polyolefins [12,13,14,15,16,17,18,19,20,21,22,23,24,25,26]. In the processing of these blends, the epoxy base, the curing agents and thermoplastic pellets or powders are usually mixed together. In general, a reaction-induced phase separation process, forming the final structure, can be observed. The processing and mixing steps are very important to obtain a good morphology and high mechanical properties. Epoxy/thermoplastic polymer blends can be produced through mechanical methods, by using batch or continuous mixers [27] or continuous polymerization reactors [28]. In the low-shear batch mixers, the thermosetting epoxy resin base is heated at high temperature to allow a better homogenization and dispersion with the thermoplastic powders, and then the curing agent is added. After degassing, the mixture can be either partially cured and then quenched or directly casted into molds for final curing. This type of mixing is not efficient for concentrations of the thermoplastic phase higher than about 30 wt %, because the viscosity of the system becomes too high to allow an efficient process. Epoxy/thermoplastic polymer blends can be also produced through non-mechanical methods, such as solvent casting, resonant acoustic mixers, and in-situ polymerization processes. Depending on the chemical nature of the constituents and on the processing route, different morphologies can be obtained for epoxy/thermoplastics blends. Homogeneous microstructures can be produced when the thermoplastic phase is soluble in the epoxy matrix and if this condition is maintained during the curing process. The homogeneity can be either due to low concentration of the thermoplastic component or due to the good affinity between the different phases. Examples of epoxy blends which remain homogeneous are those with polycarbonate and poly(ε-caprolactone) [29,30]. Heterogeneous microstructures can be obtained either with an initial immiscible mixture or starting with a homogeneous blend, which then phase separate during the curing process. Examples of epoxy blends which are heterogeneous are those with polyamides, polyvinylidene fluoride, and polybutylene terephthalate [11]. During curing, the molecular weight of the epoxy resin increases and this can lead to a phase separation between the thermoplastic phase and the epoxy matrix. If spinodal decomposition occurs, an interconnected structure is obtained, with each phase presenting a continuous three-dimensional pathway. On the other hand, if nucleation and growth occurs, an isle structure is obtained, where spheres of the minor phase are dispersed in the matrix of the major phase [3,11,31].

Poly(ε-caprolactone) (PCL) is a fossil fuel based biodegradable aliphatic polyester manufactured by ring-opening polymerization of ε-caprolactone [32]. PCL is a tough, flexible, and semicrystalline polymer with a degree of crystallinity of about 50% [33]. It presents a low melting temperature (ca. 60 °C), a low glass transition temperature (ca. 60 °C), it has a limited viscosity and it is easily processable [32]. Values of tensile strength and strain at break between 25 and 33 MPa and 450 and 1100% can be found in literature, respectively [34,35]. PCL processing techniques are commonly used for producing thermoplastic materials (i.e., injection molding, sheet extrusion and blows and slot cast film extrusion) [36]. PCL finds wide application in biodegradable packaging, in sutures, in adhesives, in non-woven fabrics or for the production of synthetic leather and dressings [32]. One of the most recent applications of PCL is as blending agent in epoxy system for self-healing applications [7,37,38].

Self-healing capability, intended as the ability to repair damages and restoring partially or completely the lost properties and functions, has been already demonstrated in many types of polymers [39] including thermoplastics, elastomers, and thermosets [40]. Generally speaking, the self-healing mechanism can be summarized in three principal steps: actuation of self-healing feature, transportation of healing agent or chemicals in the damaged zone, final chemical or physical reparation [41]. Thermoplastic polymers have demonstrated an ability to heal after heating above their melting temperature by molecular re-entanglement processes across the broken surfaces [38]. However, even if their repairing mechanism is quite easy, they were considered less interesting than thermosets when high thermal stability and solvent resistance are required. Thus, in recent years, a lot of efforts were put on the development of new healing systems for thermosetting polymers. These systems can be principally divided in two classes: intrinsic healing, where the reparation is performed by the polymer itself [40], and extrinsic healing, where an external healing agent is incorporated in the system [42]. In extrinsic healing systems the process is not completely autonomous, thus an external heating source is required to melt/soften the thermoplastic material, which then can flow, by capillary forces, toward the cracked area and repair the system. Poly(caprolactone)/epoxy blends have been studied in repairing applications with triple-shape memory effects, where the melting temperature of PCL and the glass transition temperature of the epoxy matrix have been used to fix the shape during the recovery cycle [38,43]. In a recent paper by Karger-Kocsis et al., the thermally induced healing through thermoplastic PCL dissolved in 12.5, 25, 37.5, and 50 wt %, respectively, in epoxy systems differing for their glass transition temperature (T_g_), was investigated [38]. It was found that the transition of PCL from disperse to continuous phase depends not only on the PCL amount, but also on the epoxy type and on the healing temperature.

Because of its peculiar properties, epoxy resin is the main thermosetting material used as matrix in composites, but it is quite brittle and has low fracture toughness properties. The most common solution for toughening epoxy resin is the blending with rubber elastomeric materials [44]. However, the obvious consequence is a reduction of the mechanical and thermal properties needed for high-end application such as aerospace engineering. In order to toughen and retain or even increase the thermo-mechanical properties of epoxy resin, different polymers with high molecular weight and glass transition temperatures (T_g_), like polysulfone (PSF), poly(ether sulfone) (PES), and Poly (ether imide) (PEI) were considered [3]. In these works, it has been shown that, in order to maximize the toughening effect, high performance and high molecular weight engineering thermoplastics must be used in content exceeding 20 wt %. In this way, co-continuous and phase-inverted morphologies are developed [45]. However, the increase in viscosity can lead to some problems in the processing steps. The toughening effect upon the addition of a soft rubbery phase is given by the introduction of several different mechanisms that are able to absorb energy during the fracture propagation depending on shape, size, volume fraction, surface modification, and type of the filler used. It is possible to identify in rubber reinforced materials a large number of different toughening mechanisms that can also be combined. Among the different mechanisms to toughen epoxy matrices with rigid thermoplastic particles, the most important are particle bridging, crack pinning, crack path deflection, particle-yielding-induced shear banding, particle yielding, and microcracking [46].

In the present work, a tridimesionally interconnected structure of PCL produced through filament fused fabrication (FFF) was utilized to toughen an epoxy system. FFF is a technology which allows the production of complex three-dimensional objects starting from a computer-aided design computer aided design (CAD) model. The model, in order to be recognized by the printer, is first converted into digital format and then, using a specific software, is sliced into layers [47,48,49]. Then, the model is ready for starting the printing process [50]. In this process, thermoplastic filaments are mechanically fed from a spool into an extrusion head. The extruder is heated above the characteristic critical temperature (glass transition temperature for amorphous polymers and melting temperature for semicrystalline polymers), and the softened or molten polymers are extruded through a nozzle in a x-y predefined path on a heated bed, and then solidify by cooling. After the deposition of a single layer, the heated bed moves down and the next layer is deposited, and the process continues until the final 3D object is completely produced [51,52]. The main advantages correlated to this process are its versatility, its low cost, the possibility to achieve complex geometries and to use a wide variety of polymers with multimaterial filaments with different functionalities in a single step [50]. The most used polymers for FFF technology are acrylonitrile–butadiene–styrene (ABS), poly(lactic acid) (PLA), polyamides (especially Nylon 12), polycarbonate (PC), and thermoplastic polyurethane (TPU) [53]. In a recent work of Karger-Kocsis et al., fused deposition modeling (FDM) was used to create 2D layered structure made of PCL rods that were patterned with unidirectional carbon fibers, and the resulting materials were then infiltrated with an amine-curable epoxy resin [54]. In this case, self-healing mechanism could be triggered by heat treatment above the melting temperature of the PCL phase. On the other hand, in literature, only an attempt can be found on the reinforcement of a 3D printed thermoplastic structure with a high strength resin [55], but the investigation of the toughening effect provided by the thermoplastic phase was not considered.

The aim of the present work was the development of novel thermoplastic (i.e., PCL)/thermosetting (i.e., epoxy) blends through FFF, in which the final morphology of the blend can be controlled and not solely determined by the thermodynamics of mixing of the two phases. The thermoplastic phase geometry has been designed, a priori, with a computer-aided-design software and then produced by FFF. In a second step, the PCL scaffold has been impregnated with a liquid epoxy resin. After curing of the epoxy matrix, the mechanical properties, the toughening, and the self-healing features of this novel system have been carefully analyzed and correlated with its microstructural features. In order to perform a direct comparison, also a conventional melt-mixed poly(caprolactone)/epoxy blend has been produced and characterized with the same experimental techniques.

## 2. Materials and Methods

### 2.1. Materials

In order to build the 3D printed structure, a PCL99 Naturel^®^ poly(caprolactone) filament (density = 1.145 g/cm^3^, M_n_ = 47,500 g/mol, M_w_ = 84,500 g/mol), having a diameter of 1.75 mm, was purchased from 3D4MAKERS (Haarlem, Netherlands). The epoxy base Elantech^®^ EC157 (density = 1.16 g/cm^3^, viscosity at 25 °C = 800 mPa·s) and the hardener Elantech^®^ W 342 (density = 0.96 g/cm^3^, viscosity at 25 °C = 70 mPa·s) were kindly provided by Elantas Europe Srl (Collecchio, Italy). All materials were used as received.

### 2.2. Preparation of the Samples

#### 2.2.1. 3D Printed Poly(caprolactone)-Epoxy Blends

All the PCL structures were designed with the software Solidworks^®^, provided by Dassault Systemes SE (Velizy-Villacoublay, France), and according to the model represented in Figure 1a,b. The nominal dimensions of the designed model (80 × 10 × 4 mm^3^) were selected according to ISO 179-2 standard [56]. The dimensions of the cross section were also valid for the determination of the fracture toughness according to ASTM D5045-14 standard [57] (2 < W/B < 4), where W and B are the width and thickness of the specimen, respectively. The proposed model was designed in order to allow a complete three-dimensional interconnectivity for both phases of the blend. However, when the PCL structures were put in the cavity of a silicone mold for the impregnation with the epoxy resin, it was possible to see a slight shrinkage in the mid-section of the structures. Therefore, the solution was the design of four transversal support parts in order to provide some additional stiffness. The main geometrical characteristics of the PCL structure, determined with the Solidworks^®^ software, are the following: total volume = 1119.5 mm^3^ (35.0 vol % of PCL respect to the external dimensions 80 × 10 × 4 mm^3^), volume in the span length (according to ASTM D5045 standard [57]) = 521.3 mm^3^ (32.6 vol % respect to the dimensions 40 × 10 × 4 mm^3^), amount of thermoplastic PCL in the mid cross section = 8 mm^2^ (20.0% of PCL surface content with respect to the cross-section dimensions 10 × 4 mm^2^ of the designed model).

The 3D model was designed with the software Solidworks^®^ (2013, Dassault Systemes, MA, USA) and exported in stereolithography format (STL). Through the software Slic3r, the G-code was compiled with the following printing parameters: infill percentage 100%, infill angle ± 45°, nozzle temperature = 220 °C, bed temperature = 40 °C, deposition rate = 15 mm/s. The 3D printer used to produce the samples was a Sharebot Next Generation desktop 3D printer, provided by Sharebot Srl (Nibionno, Italy), equipped with a nozzle diameter of 0.35 mm. The liquid epoxy base Elantech^®^ EC157 was degassed for 20 min at room temperature, and then the hardener Elantech^®^ W342 was added and manually mixed for 2 min. The mixture was degassed for 5 min at room temperature, and subsequently cast into the 3D printed PCL structure, that was previously placed inside a silicone mold. The resulting blend was then subjected to the curing procedure at room temperature for 24 h, followed by the postcuring at 60 °C for 15 h. Therefore, in this work neat epoxy samples (EP), neat PCL filament (i.e., PCL(FIL)) and the corresponding 3D printed structure (i.e., PCL(3D)) were considered for the preparation of 3D printed poly(caprolactone)/epoxy blends (i.e., EP-PCL(3D)), in which the PCL content was kept constant at 35 vol %. In Figure 2 are reported representative images of the prepared PCL(3D), EP, and EP-PCL(3D) samples.

#### 2.2.2. Melt-Mixed Poly(caprolactone)-Epoxy Blends

A blend of PCL and epoxy resin was prepared with a PCL content equal to 20.0 wt %. The selected amount of PCL was the same present in the mid cross section of the produced EP-PCL(3D) blends. Small PCL pieces were prepared by cutting the corresponding filaments, and then they were added to the epoxy base Elantech^®^ EC157 in a steel container connected to a water circuit. The temperature of the water was set to 85 °C and PCL was melt-mixed with the liquid epoxy base by mechanical stirring at 750 rpm for 90 min with a mechanical mixer. The mixture was then degassed for 30 min at room temperature. The temperature was then raised at 70 °C and the hardener Elantech^®^ W342 was added and manually mixed for 2 min. The mixtures were then immediately casted into a silicone molds, and degassed again. After performing the same curing procedure reported for the 3D printed PCL epoxy blends, 80 × 10 × 4 mm^3^ specimens were obtained. The acronym EP-PCL was utilized to design these samples.

### 2.3. Experimental Techniques

#### 2.3.1. Characterization of the Constituents

The elastic modulus of the blend constituents (i.e., epoxy and PCL 3D printed structure) was evaluated with quasi-static tensile tests performed using an Instron^®^ 5969 testing machine (Instron; Norwood, MA, USA) equipped with a load cell of 50 kN. According to ISO 527 standard [58] 1BA samples were produced by 3D printing for PCL and by casting for EP. The crosshead speed was set equal to 0.25 mm/min and the strain was measured with an extensometer Instron^®^ 2620 (Instron; Norwood, MA, USA), with a gauge length of 12.5 mm. The maximum deformation reached in all the tests was limited to 1% and the elastic modulus was evaluated with the method of the secant line between deformation levels of 0.05% and 0.25%. Failure properties of the constituents were measured with the same machine and the crosshead speed was set equal to 100 mm/min and the strain values were obtained by simply referring to the crossbar displacement. Instead, for EP specimens, the crosshead speed was set equal to 1 mm/min. The strain at break was calculated normalizing the elongation at break with the initial distance between the grips, set at 58 mm. Five specimens were tested for both PCL(3D) and EP samples.

#### 2.3.2. Evaluation of the Flexural and Fracture Behavior

Three-point bending flexural tests were performed according to American Society for Testing and Materials (ASTM) D790 standard [59], with an Instron 5969 test machine (Instron; Norwood, MA, USA) with a load cell of 50 kN. The nominal dimensions of the tested EP, EP-PCL(3D), and EP-PCL specimens were 80 × 10 × 4 mm^3^. The span length was set to 65 mm and the crosshead speed was fixed at 1.68 mm/min. At least five specimens were tested for each sample. In this way, the tangent modulus of elasticity (E_B_), the flexural stress at break (σ_fB_) and the flexural strain at break (ε_fB_) were determined.

For as concerns the evaluation of fracture behavior under quasi-static conditions, single edge notched bending (SENB) specimens with nominal dimensions 80 × 10 × 4 mm^3^ were tested, for EP, EP-PCL(3D), and EP-PCL blends, in three-point bending mode by using a universal Instron^®^ 5969 testing machine, equipped with a load cell of 50 kN. The measurements were carried out at room temperature using a span length of 40 mm and a crosshead speed of 10 mm/min, according to the ASTM D5045 standard. Before testing, the samples were sharply notched with a razor blade (notch radius of 0.01 mm) and the notch depth (a) was one half of the sample width. In this way, it was possible to determine the critical stress intensity factor (K_IC_) of the prepared samples, with the expressions reported in Equations (1) and (2)
(1)KQ=PQB W12 f(x)
(2)f(x)=6x12 1.99−x(1−x)(2.15−3.93x+2.7x2)(1+2x)(1−x)32
where K_Q_ is the tentative value for K_IC_, P_Q_ is the tentative value for the load and it could be equal or lower to P_max_, that is the maximum load sustained by the samples, B and W are respectively the thickness and the width of the samples, and f(x) is a calibration factor, with x = a/W. Five specimens were tested for EP, EP-PCL(3D), and EP-PCL blends, and the linearity and plasticization validity criteria required by the ASTM D5045 were carefully checked.

For EP, EP-PCL(3D), and EP-PCL blends, a new set of SENB specimens, with the same dimensions reported for fracture tests under quasi-static conditions, was tested under impact conditions. The measurements were performed using a Charpy impact machine provided by CEAST, and the load–displacement curves were recorded using a tup extensometer in the hammer. A mass of the hammer equal to 2.5 kg, a starting angle of 32°, an impact speed of 1 m/s and a span length of 40 mm were utilized to test the samples. The procedure for the determination of the fracture toughness was set according to [60], which was a test protocol used for the compilation of the ISO 17281 standard. In this way, K_IC_ and specific energy absorbed under impact conditions (U_spec_), computed as the total impact energy divided by the resisting cross section of the material, were determined. Even in this case, five specimens were tested for each sample. Three-point bending configurations was selected to evaluate the toughness of the studied materials under quasi-static conditions, because it resembled the Charpy impact test to study the fracture toughness at higher speed.

#### 2.3.3. Evaluation of the Healing Behavior

EP, EP-PCL(3D), and EP-PCL blends samples, tested and fractured both in quasi-static mode and under impact conditions, were then healed by holding the two broken halves under pressure (7 MPa) in an oven at T = 80 °C for 30 min, by using a home-made screw device. The above-mentioned healing temperatures and time were selected according to the parameters reported in literature [38]. The healed samples were re-tested following the same procedure reported for the evaluation of the fracture behavior in quasi-static and impact mode, and the healing efficiency (*η_healing_*) was determined according to Equation (3)
(3)$$\eta_{healing}=\frac{K_{IC,\;\;healed}}{K_{IC,\;\;virgin}}$$
where *K_Ic,healed_* is the fracture toughness after healing, and *K_Ic,virgin_* is the fracture toughness before healing.

#### 2.3.4. Microstructrural Characterization

The top view and the fracture surfaces of EP-PCL(3D) blend specimens, tested under impact conditions, were observed, before and after healing, through a Zeiss Axiophot optical microscope, equipped with a Leica DC300 digital camera. Microstructural observations of the fracture surfaces of EP-PCL blend were performed by a Zeiss Supra 40 high resolution Scanning Electron Microscope (SEM) operating at an accelerating voltage of 2.5 kV, after that a platinum palladium conductive coating was applied on the samples.

## 3. Results and Discussion

### 3.1. Characterization of the Constituents

Density measurements on the EP, PCL(FIL), and PCL(3D) samples (not reported for the sake of brevity) demonstrate that the values obtained for the two base components (EP and PCL(FIL)) are compatible with their respective datasheets. Moreover, the density of the PCL(3D) sample (1.126 g/cm^3^) is the 98.3% of that of PCL(FIL) sample, meaning that the amount of voids in the specimens produced by 3D printing is very limited.

From differential scanning calorimetry (DSC) measurements on the EP sample (not reported for the sake of brevity) it was demonstrated that the glass transition temperature (Tg) of cured EP measured during the first heating scan (i.e., 85.4 °C), is in accordance with the value reported in its technical data sheet, and thus cured EP undergoes to an almost complete curing process. The higher glass transition temperature values reported for the EP sample in the second heating scan (106.7 °C), are compatible with the values achievable with the second curing method, as suggested in its technical data sheet. Moreover, DSC tests on PCL(FIL) and PCL(3D) samples demonstrate that the two PCL types show comparable thermal properties in terms of melting temperature (Tm = 61 °C), glass transition temperature (Tg = −69 °C) and degree of crystallinity (Χc = 56%)). The above-mentioned thermal properties are in accordance with the literature data [61,62], and confirm that the 3D printing process does not substantially affect the thermal properties of the PCL filaments.

In Figure 3a, b, representative stress–strain curves obtained from quasi-static tensile tests at break on the EP and PCL(3D) constituents are respectively reported, while in Table 1 the most important results are summarized. The stress–strain curve of the EP sample is a typical representation of a brittle material without any plasticization, and the failure properties collected in Table 1 are compatible with those provided by the producer of the resin. In the tensile tests on PCL samples, once necking starts to appear in the mid cross-section, it propagates along the entire length of the dumbbell specimens, reaching elevated strain values. The material starts yielding around 16–17 MPa and once the necking begins, there is a drop in the stress to about 12 MPa. Then, the plastic deformation propagates maintaining a constant stress, up to a strain of 250%. In Figure 3b the stress increase in the curve at high deformation level, with the corresponding fluctuations, is due to the plastic deformation of the handles of the dumbbell specimens. The stress–strain curve and the relative values at yield and at break find correspondence in literature data [35,36]. Comparing tensile properties results reported in Table 1, it is evident that both the stiffness and the stress at break (σb) of EP samples are considerably higher with respect to that of the 3D printed PCL samples, while the strain at break (εb) values are much lower. This difference can play a crucial role in the healing capability of the prepared blends and in their fracture behavior.

### 3.2. Evaluation of the Flexural and Fracture Behavior

In Table 2 the results of quasi-static flexural tests on neat epoxy, EP-PCL(3D) and EP-PCL samples are compared. The stress–strain curve of EP-PCL(3D) (not reported for the sake of brevity), is similar to that of the EP sample, and shows a typical brittle behavior with absence of plasticity. EP-PCL(3D) blend presents a strain at break similar to EP, so PCL introduction does not lead to any increase in ductility. The main reason behind this behavior can be found in the complete separation of the two phases. In fact, when EP breaks, the stresses reached during the testing are so high that PCL cannot withstand them and, in turn, suddenly breaks. Instead, as natural consequence of the addition of PCL, EP-PCL(3D) flexural modulus and stress at break are noticeably reduced. However, even though the mechanical properties have been decreased, they are still suitable for the greatest part of the applications in which epoxy resins are utilized. The increased plasticity in the EP-PCL sample is reflected in the much higher strain at break value, which has been increased of ~75% with respect to EP-PCL(3D). In the melt-mixing process, the resulting morphology and the good dispersion of the PCL phase allows the toughening phase to start yielding at lower values of stress, consequently reaching higher values of strain at break. Considering the flexural modulus and stress at break values, even though EP-PCL(3D) has almost the double of the PCL weight content with respect to EP-PCL samples, the mechanical properties are slightly higher. The reason can be found in the fact that in EP-PCL(3D) the two base components are completely separated and behave in an independent way, and so the properties of epoxy resin are more retained.

In Figure 4a representative load–displacement curves of EP, EP-PCL(3D), and EP-PCL samples obtained from three-point bending tests for fracture toughness evaluation in quasi-static conditions are reported, while in Figure 5a the values of the critical stress intensity factor (K_IC_) are collected. From the load–displacement curve of the EP sample it is possible to see that this material completely fulfills the linearity requirements reported in the ASTM D5045 standard. Moreover, with the obtained value of fracture toughness (i.e., 1.03 ± 0.07 MPa·m^1/2^), EP sample also satisfies the plasticity verification. The reported K_IC_ value lays in the typical range of epoxy based materials (0.4–2.2 MPa·m^1/2^) [63]. From the load–displacement curve of EP-PCL(3D) sample obtained in quasi-static tests it can be seen that EP and PCL phases behave in an independent manner. In fact, after the failure of EP structure, PCL starts yielding maintaining a constant load and keeping the two broken halves together. Thus, if the force at break of the EP phase is selected for the determination of the fracture toughness, it is possible to satisfy both the linearity and plasticity verifications. The addition of the PCL(3D) phase within the epoxy and the creation of the designed 3D printed model has a beneficial effect on the fracture toughness of the material, because EP-PCL(3D) sample shows higher K_IC_ values with respect to the EP sample. From the representative load–displacement curve of EP-PCL sample it is possible to notice that in this blend the main components do not behave in a completely independent manner. In fact, the morphology achieved in melt-mixing operations allows to obtain a higher plasticity before the failure with respect to EP and EP-PCL(3D) samples and it gives the possibility to activate several different toughening mechanisms related to the presence of a rubbery phase dispersed in a brittle matrix. In particular, among them the most significant involve rubber phase debonding and cavitation, localized shear banding of matrix, and the rubber particle bridging and tear that pushes the crack to be deflected or to be pinned and stopped [64,65]. The mean P_MAX_/P_Q_ value is 1.16, thus an apparent fracture toughness must be considered as result. The K_IC_ value reported for this material (1.28 ± 0.12 MPa·m^1/2^) demonstrates a higher toughening effect with respect to the EP-PCL(3D) sample.

It could be also interesting to evaluate the fracture behavior under impact conditions. In Figure 4b, representative load–displacement curves of EP, EP-PCL(3D), and EP-PCL samples obtained from three-point bending tests on notched specimens under impact conditions are collected, while in Figure 5b the resulting K_IC_ values are summarized. The load–displacement curve reported for the EP sample is typical for a brittle material, with the load that suddenly drops to zero after the failure. The tested EP specimens fulfil the linearity and plasticity requirements, with a P_MAX_/P_Q_ values equal to 1.10. The value of fracture toughness is similar to that obtained in quasi-static tests. Considering the EP-PCL(3D) sample, after the failure of EP part the load does not automatically drop to zero neither under impact conditions. EP-PCL(3D) blend is able to absorb more energy and can sustain a significant load for higher displacement values. This behavior is the same observed in quasi-static tests, thus it is selected the force at break of EP as P_Q_ value for the determination of the fracture toughness, and the linearity and plasticity verification are completely satisfied. From the values reported in Figure 5b, it can be seen that the K_IC_ of EP-PCL(3D) sample is similar to that obtained in quasi-static tests, thus a slightly toughening effect with respect to neat EP can be achieved. From the load–displacement curve obtained under impact conditions for the EP-PCL sample it is possible to notice that the produced blend leads to a toughening effect with respect to the EP sample. The determined K_IC_ value is similar to that obtained in quasi static conditions, thus it is slightly higher with respect to that of EP-PCL(3D) sample. However, the registered mean P_MAX_/P_Q_ value (> 1.25) suggests that a certain plasticization occurred near the crack tip. This phenomenon was highlighted by the presence of craze formation in EP-PCL composite according to SEM observation that will be shown later in the manuscript. It is also interesting to evaluate the effect of the PCL phase on the epoxy considering the specific energy values absorbed under impact conditions (Uspec). The best expression of the independent behavior of the two components inside the EP-PCL(3D) blend can be seen in Figure 5b. In fact, thanks to the tough PCL(3D) phase, the specific energy is increased more than four times with respect to EP (2.75 kJ/m^2^) due to the higher strain at break of the PCL. Also EP-PCL shows an increased specific energy with respect to the EP sample, with a relative increase of more than 50%, but the value is lower than that obtained for the EP-PCL(3D) samples (1.74 kJ/m^2^). Thus, it can be concluded that the presence of 3D printed PCL as self-standing phase within the epoxy matrix leads to a considerable increase in impact properties with respect to the conventional production technique (i.e., melt mixing).

### 3.3. Evaluation of the Thermal Healing Behavior

In Figure 6a representative load–displacement curves of EP-PCL(3D) samples (healed and not healed) obtained in fracture toughness tests under quasi-static conditions are compared, while in Table 3 the healing efficiency values are collected. Healed EP-PCL(3D) sample shows a linear behavior, with a corresponding mean P_MAX_/P_Q_ value equal to 1.15. Therefore, the calculated fracture toughness has to be considered as an apparent value. Even if the healing efficiency obtained in quasi-static tests is only 15%, this result is in accordance with the values reported in literature for similar mid-cross section PCL volume fractions [38]. In fact, despite in the present work a higher pressure was applied to heal the samples (i.e., 7 MPa), all the other parameters (i.e., the healing temperature and time) are the same [38]. Finally, since the healing efficiency of EP is practically 0% and the two phases are completely independent in the designed 3D structure, only PCL is considered able to repair during the healing procedure and so the final mechanical properties of the healed objects are only related to the PCL phase. Analogous results can be found in the work of Szebényi et al., where the curve after healing of an epoxy/carbon fibers/PCL system in a quasi-static three-point bending test was similar to that of the neat PCL, due to the fact that the reinforcing epoxy layers lost their mechanical efficiency, being fully fractured [54].

In Figure 6b representative load–displacement curves of EP-PCL samples (healed and not healed) obtained in fracture toughness tests under quasi-static conditions are compared. The tested EP-PCL specimen fulfills the linearity and plasticity requirements, with a mean P_MAX_/P_Q_ value equal to 1.10. The reported value of healing efficiency (see Table 3) is in accordance both with that obtained for EP-PCL(3D), and also with those reported in literature [38,66]. In the case of EP-PCL blends produced with melt-mixing, higher healing efficiency can be obtained when the morphologies of the two phases are completely co-continuous, or even better efficiencies can be achieved with a phase-inverted morphology at high PCL content. Another important aspect for the determination of self-healing performances is the type of selected epoxy resin matrix. In fact, for similar PCL volume fractions, higher values of efficiency have been reported using either much lower or much higher glass transition temperature epoxies [38].

The representative load–displacement curves of healed and not healed EP-PCL(3D) samples, obtained under impact conditions, are reported in Figure 7a. In this case, the healing efficiency, as shown in Table 3, is higher with respect to the value obtained in quasi static conditions and the reason can be found looking at the curve of Figure 7a. In fact, the linear behavior is not satisfied, with a corresponding mean P_MAX_/P_Q_ value higher than 1.20. Moreover, the fracture loads of healed EP-PCL(3D) samples are very low, and in many cases it is not possible to distinguish them with adequate accuracy, due to the inevitable inertial peak typical of impact tests. In this sense, it should be better to refer to quasi-static test results for a more precise analysis of the healing efficiency. In Figure 7b the representative load–displacement curves of healed and not healed EP-PCL(3D) samples are compared, obtained under Charpy impact conditions. As shown in Table 3, also in this case the healing efficiency is higher with respect to the value obtained in quasi static conditions, and the reasons are the same reported PCL(3D) samples. Finally, because the same behavior is reported for two materials based on different production processes, it can be concluded that the testing of very weak interfaces is not a feasible option when impact conditions are involved.

### 3.4. Microstructural Behavior

The optical microscope images of the fracture surfaces of EP-PCL(3D) specimens, tested under impact conditions, are respectively shown, before and after healing, in Figure 8a, b. In both images it is possible to see on the right the rough surface produced by the razor blade during the notching of SENB specimens, and on the left the smooth fracture surface due to the fracture propagation during the tests. The observed fracture propagation profile is a proof of the independent behavior between the two phases (i.e., epoxy and PCL). In fact, the crack starts to propagate within the epoxy phase producing a very smooth profile, typical of a brittle material. Instead, PCL phase acts as an obstacle to the fracture propagation, and this behavior can be clearly seen in the elevated value of the absorbed impact energy. On the contrary, the fracture toughness of this material mainly depends by the maximum load sustained by the epoxy phase. In both figures, it is also evident how the cross section of the 3D printed PCL domains is not perfectly square (as shown in Figure 1a,b), probably because of some difficulties in the printing process and the partial softening the PCL phase in the postcuring operations. Comparing the micrographs before and after the healing process, no significant microstructural differences due to the healing process can be detected.

SEM analysis was carried out in order to compare the different morphologies of the fracture surfaces of EP-PCL samples. The micrographs of the fracture surfaces of the SENB samples tested in quasi-static and impact conditions before the healing procedure are reported in Figure 9a,b, respectively. As it is possible to see in Figure 9a, the fracture surfaces obtained in quasi-static conditions shows an evident surface damage with an extensive deformation of the PCL phase, that probably acts both as a stand-alone continuous phase and also as phase dispersed in the epoxy resin matrix. This craze formation observed on the fracture surface is initiated by void formation that follows the interfacial debonding between PCL phases and epoxy matrix, so the energy dissipated by the craze formation when PCL is finely dispersed is greater than that in neat epoxy regions that surround the PCL 3D printed strands. This result is consistent with the information found in literature for epoxy/PCL blends [38,66].

In the SENB sample tested under impact (see Figure 8b), it is possible to see that, due to the higher testing rates, the deformation of the PCL phase is much less evident. This could also justify the conclusions reported in the analysis of the optical microscope image reported in Figure 8a.

It could be also interesting to compare the microstructural features of the EP-PCL samples before and after the healing process. Therefore, in Figure 10 SEM micrographs of fracture surfaces of EP-PCL samples tested in quasi-static and impact mode, before and after the healing process, are compared. Regardless the testing speed, the fracture surfaces after healing present a much smoother surface. The smoothed profile can probably be ascribed to the combined action of the temperature and the pressure during the thermal mending, that causes the melting of the PCL phase and its diffusion within the fracture profile. Nevertheless, considering that the mechanical properties of the PCL are much lower to those of the epoxy, the healing efficiency of the prepared EP-PCL system is rather limited. Further efforts will be thus performed in the future to detect a thermoplastic healing agent with mechanical properties matching those of the epoxy.

## 4. Conclusions

From quasi-static and impact tests for fracture toughness evaluation, EP-PCL sample showed a toughening effect with respect to the neat epoxy, probably because of the homogenous dispersion of the PCL domains with the thermosetting phase. In the EP-PCL(3D) system, the two phases behaved in an independent way, and the fracture propagation of the epoxy constituent was followed by the progressive yielding of the PCL phase. Therefore, the fracture toughness of this samples was slightly lower with respect to that observed for the EP-PCL system, but the toughening effect was still present. Interestingly, impact tests highlighted that, even if the impact energy detected for the EP-PCL sample was much higher than that obtained for the neat epoxy, it was more than 50% lower with respect to that shown by the EP-PCL(3D) blend. The phase separation during the tests allowed the tough PCL phase to express its full potential in impact energy absorption. On the other hand, the healing efficiency values of the two blends were rather low, even if in accordance with those found in literature. Optical microscope images on the fracture surfaces of SENB samples highlighted the presence of two self-standing phases in EP-PCL(3D) samples. In these samples, the cracking started and propagated in the epoxy phase, while the PCL phase contribute was more related to the energy absorption though plastic deformation. From SEM analysis of fracture surfaces of EP-PCL blend it was shown that PCL deformation and damaging was less evident at high loading rates, while the healing process rendered the fracture surface smoother, because of the melting of the PCL phase.

From a comparison with the properties of the blends produced through conventional melt mixing, it can be stated that the most interesting features of the 3D printed PCL/epoxy blends were mainly related to their good combination of mechanical, impact, and toughening properties. The main advantage seemed to be represented by the possibility to tune the final properties thanks to the designed model and to the completely self-standing behavior of the two phases.

## Figures and Tables

**Figure 1 materials-13-00819-f001:**
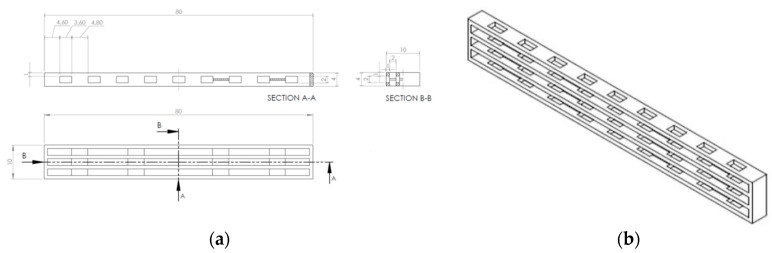
(**a**) Technical drawing and (**b**) 3D representation of the PCL model.

**Figure 2 materials-13-00819-f002:**
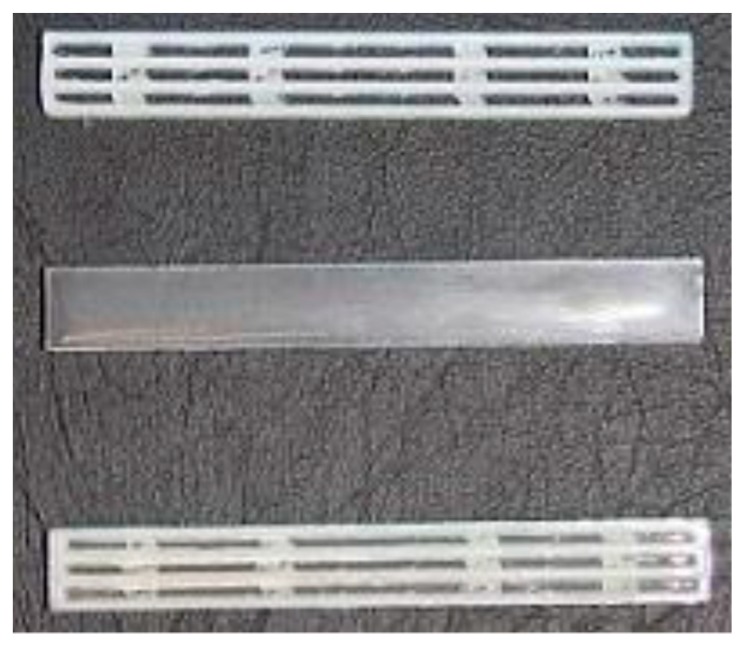
Representative images of the prepared PCL(3D), EP, and EP-PCL(3D) samples.

**Figure 3 materials-13-00819-f003:**
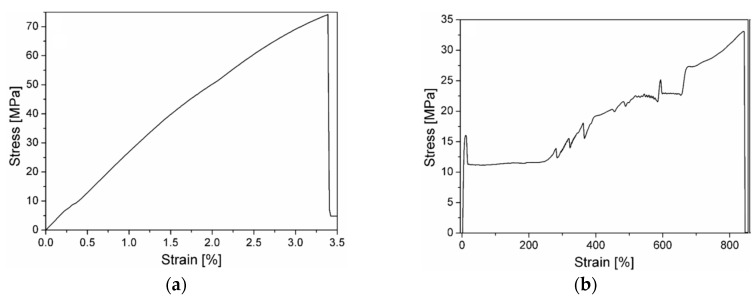
Representative stress–strain curves obtained from quasi-static tensile tests at break on the constituents. (**a**) EP, (**b**) PCL(3D) samples.

**Figure 4 materials-13-00819-f004:**
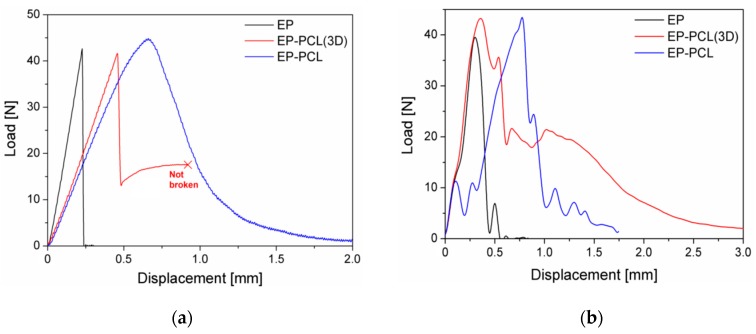
Representative load–displacement curves of EP, EP-PCL(3D) and EP-PCL samples obtained from three-point bending tests for fracture toughness evaluation. (**a**) Quasi-static conditions, (**b**) impact conditions.

**Figure 5 materials-13-00819-f005:**
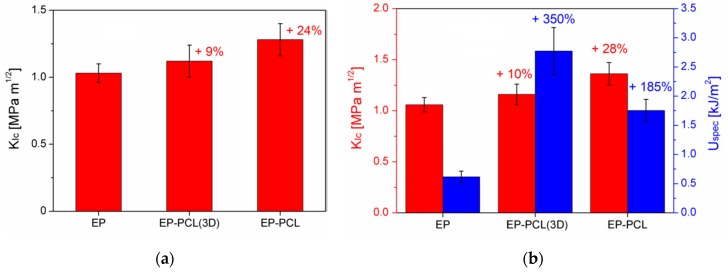
Critical stress intensity factor (K_IC_) and specific absorbed impact energy (U_spec_) of EP, EP-PCL(3D), and EP-PCL samples obtained from three-point bending tests on notched specimens. (**a**) Quasi-static conditions, (**b**) impact conditions.

**Figure 6 materials-13-00819-f006:**
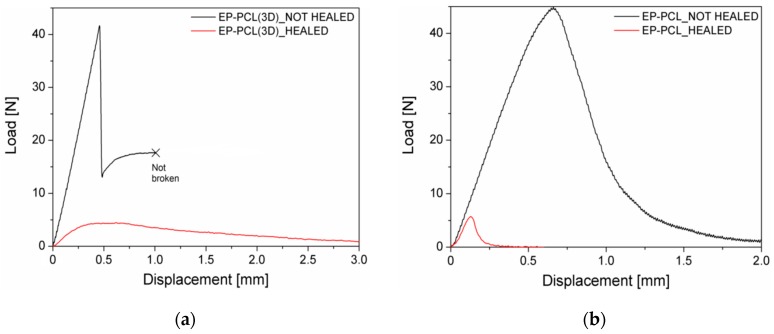
Representative load–displacement curves of (**a**) EP-PCL(3D) and (**b**) EP-PCL samples (healed and not healed) obtained in fracture toughness tests under quasi-static conditions.

**Figure 7 materials-13-00819-f007:**
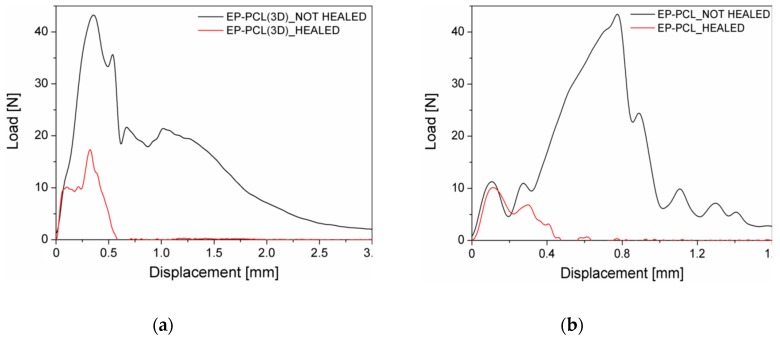
Representative load–displacement curves of (**a**) EP-PCL(3D) and (**b**) EP-PCL samples (healed and not healed) obtained in fracture toughness tests under impact conditions.

**Figure 8 materials-13-00819-f008:**
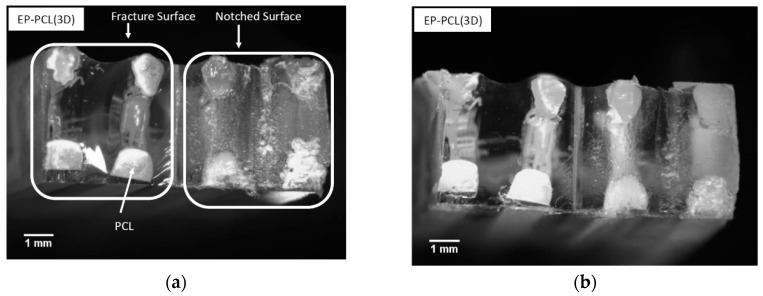
Optical microscope images of the fracture surface of an EP-PCL(3D) specimen tested under impact conditions. (**a**) Before healing and (**b**) after healing.

**Figure 9 materials-13-00819-f009:**
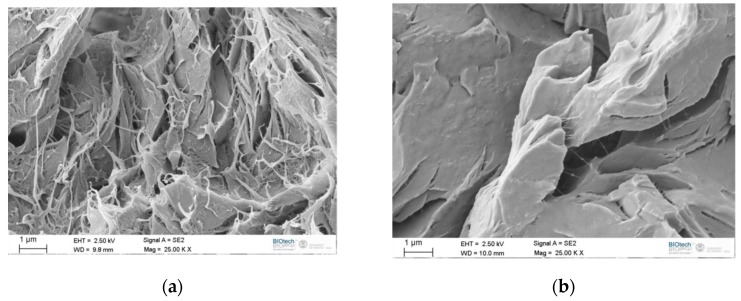
SEM micrographs of fracture surface of EP-PCL samples before the healing. (**a**) Quasi-static tests and (**b**) impact tests.

**Figure 10 materials-13-00819-f010:**
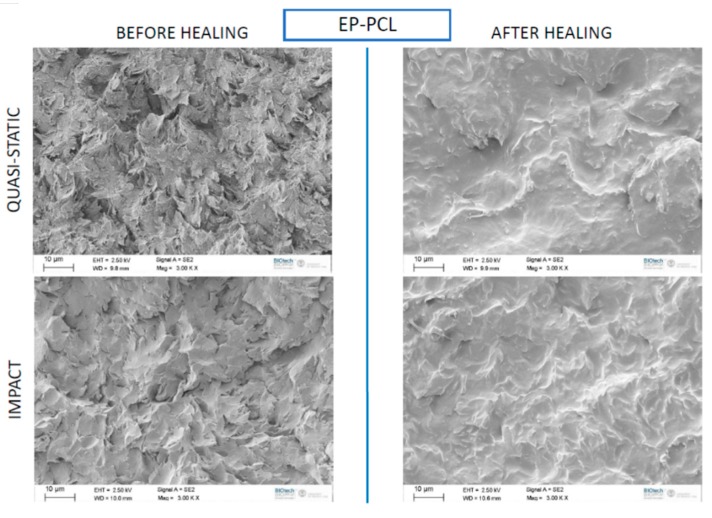
SEM micrographs of fracture surfaces of EP-PCL samples before and after the healing process.

**Table 1 materials-13-00819-t001:** Results of quasi-static tensile tests on EP and PCL(3D) samples.

Sample	E (MPa)	σ_y_ (MPa)	σ_b_ (MPa)	ε_b_ (%)
EP	2749 ± 39	-	69.4 ± 7.7	3.1 ± 0.5
PCL(3D)	421 ± 12	16.6 ± 0.5	33.7 ± 2.9	853.0 ± 25.0

**Table 2 materials-13-00819-t002:** Results of quasi-static flexural tests on EP, EP-PCL(3D), and EP-PCL samples.

Sample	E_B_ (MPa)	σ_fB_ (MPa)	ε_fB_ (%)
EP	3070 ± 99	108.0 ± 5.0	4.0 ± 0.2
EP-PCL(3D)	1709 ± 109	55.8 ± 6.3	3.6 ± 0.3
EP-PCL	1541 ± 24	47.1 ± 1.5	6.2 ± 0.5

**Table 3 materials-13-00819-t003:** Healing efficiency values of EP-PCL(3D) and EP-PCL samples under quasi-static and impact conditions.

Sample	K_IC,v_ (MPa·m^0.5^)	K_IC,h_ (MPa·m^0.5^)	η_h_ (%)
Quasi-static tests			
EP-PCL(3D)	1.12 ± 0.12	0.16 ± 0.02	15 ± 2
EP-PCL	1.28 ± 0.12	0.23 ± 0.07	18 ± 5
Impact tests			
EP-PCL(3D)	1.15 ± 0.10	0.39 ± 0.11	34 ± 9
EP-PCL	1.35 ± 0.11	0.36 ± 0.07	27 ± 6

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
