# Peer review of "Novel Poly(Caprolactone)/Epoxy Blends by Additive Manufacturing"

_materials, 2020, doi:10.3390/ma13040819_

Round 1

Reviewer 1 Report

The present work develops a novel PCL reinforced composite materials with epoxy as the matrix. Mechanical properties, self-healing, and thermal properties are improved due to the addition of PCL fabricated by FFF method. The introduction is very comprehensive with detailed overview on the relevant field and progress. Sufficient literature has been reviewed. The results are well discussed and presented. 

One recommendation for the authors is to add a brief discussion why three points bending is selected to characterise the toughness and fracture energy of the sample materials. Are those method better simulating the real-life application of the sample materials? A number of testing protocols can be utilized to measure the fracture energy such as tearing test, impact and pure shear. 

Another recommendation is to give more details on the synergistic effect of PCL and epoxy to toughen the composite. This will be a very interesting part. I saw some good discussions on page 9, and this part can be further developed regarding to how the fracture is dissipated by PCL chains or if there is any damaged zone ahead of the crack tip. 

Author Response

Reviewer 1

Q. The present work develops a novel PCL reinforced composite materials with epoxy as the matrix. Mechanical properties, self-healing, and thermal properties are improved due to the addition of PCL fabricated by FFF method. The introduction is very comprehensive with detailed overview on the relevant field and progress. Sufficient literature has been reviewed. The results are well discussed and presented. 

R. We thank the reviewer for the time dedicated to our manuscript and for the precious indications.

Q. One recommendation for the authors is to add a brief discussion why three points bending is selected to characterise the toughness and fracture energy of the sample materials. Are those method better simulating the real-life application of the sample materials? A number of testing protocols can be utilized to measure the fracture energy such as tearing test, impact and pure shear. 

R. Three-point bending configurations was selected to evaluate the toughness of the studied materials under quasi-static conditions because it gives a solid analysis of the fracture toughness in case of brittle materials and also because it resembled the Charpy impact test to study the fracture parameters at higher speed. A sentence is added to our manuscript and it is highlighted in red.

Q. Another recommendation is to give more details on the synergistic effect of PCL and epoxy to toughen the composite. This will be a very interesting part. I saw some good discussions on page 9, and this part can be further developed regarding to how the fracture is dissipated by PCL chains or if there is any damaged zone ahead of the crack tip. 

R. As highlighted inside our manuscript, the fracture energy is consumed in different ways depending if PCL is dispersed in the EP matrix or it is organized in 3D printed strands. In the first case, several different toughening mechanisms related to the presence of a rubbery phase dispersed in a brittle matrix are present according to SEM observation such as craze formation. In the latter case, upon the fracture of the matrix, 3D printed strands of PCL are able to keep the two broken halves together and that results in a larger increase of the absorbed fracture energy. Some comments and some more references are added to our manuscripts and they are highlighted in red.

Reviewer 2 Report

The healing efficiency values of the two blends were rather low. Nevertheless, the physical mechanism of healing should be more highlighted/compared in the  case of the evaluated material systems.

Author Response

Reviewer 2

Q. The healing efficiency values of the two blends were rather low. Nevertheless, the physical mechanism of healing should be more highlighted/compared in the case of the evaluated material systems.

R. We thank the reviewer for the time dedicated to our manuscript. The physical mechanism of healing in the case of the evaluated material is the molecular re-entanglement across the broken surfaces triggered by heat. A sentence highlighted in red that emphasize this feature is added to our manuscript.
